# Autoregulatory control of microtubule binding in doublecortin-like kinase 1

**Regina L Agulto[1], Melissa M Rogers[1], Tracy C Tan[1], Amrita Ramkumar[1†], Ashlyn M Downing[1‡], Hannah Bodin[1], Julia Castro[1], Dan W Nowakowski[2], Kassandra M Ori-McKenney[1]\***

[1]Department of Molecular and Cellular Biology, University of California, Davis, Davis, United States; [2]N Molecular Systems, Inc, Palo Alto, United States

**Abstract** The microtubule-associated protein, doublecortin-like kinase 1 (DCLK1), is highly expressed in a range of cancers and is a prominent therapeutic target for kinase inhibitors. The physiological roles of DCLK1 kinase activity and how it is regulated remain elusive. Here, we analyze the role of mammalian DCLK1 kinase activity in regulating microtubule binding. We found that DCLK1 autophosphorylates a residue within its C-terminal tail to restrict its kinase activity and prevent aberrant hyperphosphorylation within its microtubule-binding domain. Removal of the C-terminal tail or mutation of this residue causes an increase in phosphorylation within the doublecortin domains, which abolishes microtubule binding. Therefore, autophosphorylation at specific sites within DCLK1 has diametric effects on the molecule's association with microtubules. Our results suggest a mechanism by which DCLK1 modulates its kinase activity to tune its microtubule-binding affinity. These results provide molecular insights for future therapeutic efforts related to DCLK1's role in cancer development and progression.

**\*For correspondence:**
kmorimckenney@ucdavis.edu

**Present address:** †Synthego Corporation, Suite C, Redwood City, United States; ‡Ayni Therapeutics, Inc, Santa Cruz, United States

## Introduction

Growth is an essential process of life. Unchecked cellular growth, however, is a hallmark of cancer. Therefore, the process of cell division is often a target of cancer therapeutics (*Steinmetz and Prota, 2018*; *Wieczorek et al., 2016*). The macromolecular machine responsible for accurately segregating chromosomes during eukaryotic cell division is the bipolar mitotic spindle, a structure composed of dynamic microtubules organized by a multitude of microtubule-associated proteins (MAPs) (*Hornick et al., 2010*; *Barisic and Maiato, 2016*). Doublecortin-like kinase 1 (DCLK1), formerly known as DCAMKL1 and KIAA0369, is one such MAP that is also upregulated in a range of cancers, such as pancreatic, breast, bladder, colorectal, gastric, and hepatocellular carcinoma (*Burgess et al., 1999*; *Lin et al., 2000*; *Li and Bellows, 2013*; *Meng et al., 2013*; *Qu et al., 2015*; *Liu et al., 2016*; *Fan et al., 2017*; *Kadletz et al., 2017*; *Jiang et al., 2018*; *Zhang et al., 2017*). As a consequence, many studies have focused on developing small-molecule inhibitors against DCLK1 kinase activity in an effort to control cancer growth (*Westphalen et al., 2017*; *Weygant et al., 2014*; *Ferguson et al., 2020*). However, it is currently unclear if DCLK1 kinase activity, microtubule-binding activity, or both are involved in the molecule's functions during cell division. Therefore, a mechanistic understanding of DCLK1, both at the molecular and biological level, is currently lacking.

DCLK1 is a member of the doublecortin (DCX) superfamily, which also includes DCX, DCDC2, and retinitis pigmentosa 1 (RP1), all of which are implicated in human disease (*Westphalen et al., 2017*; *Reiner et al., 2006*; *Sullivan et al., 1999*; *Meng et al., 2005*; *Gleeson et al., 1998*; *Francis et al., 1999*). At its N-terminus, DCLK1 contains two tandem DCX domains (DC1 or N-DC: aa 54–152 and DC2 or C-DC: aa 180–263) (*Figure 1A*), which are highly conserved among other family members (*Lin et al., 2000*; *Reiner et al., 2006*; *Sapir et al., 2000*; *Kim et al., 2003a*; *Taylor et al., 2000*). DCLK1 and its paralog, DCX, were originally identified and characterized for

their functions during neuronal development, including neurogenesis and neuronal migration (*Burgess et al., 1999*; *Gleeson et al., 1998*; *Francis et al., 1999*; *Bai et al., 2003*; *Burgess and Reiner, 2000*; *Jean et al., 2012*). Although the roles of DCLK1 and DCX in neurodevelopment have been phenotypically described in vivo, the molecular basis for these observations remains ill-defined. Prior studies have shown that DCLK1 and DCX may act as microtubule stabilizers, nucleators, and regulators of microtubule-based transport (*Liu et al., 2012*; *Moores et al., 2004*; *Moores et al., 2006*; *Bechstedt and Brouhard, 2012*; *Bechstedt et al., 2014*; *Lipka et al., 2016*; *Monroy et al., 2020*; *Ettinger et al., 2016*; *Patel et al., 2016*). Dissecting the mechanisms by which DCLK1 binds to the microtubule can therefore provide insights into the microtubule-binding behaviors of other DCX family members and how they may be subverted in disease.

The C-terminal portion of DCLK1 contains a serine/threonine kinase domain and an unstructured C-terminal tail that shares sequence similarities with calcium/calmodulin-dependent protein kinase I (CaMKI) (*Shang et al., 2003*; *Edelman et al., 2005*). For both DCLK1 and CaMKI, removal of a distal C-terminal 'tail' region results in an increase in kinase activity (*Patel et al., 2016*; *Shang et al., 2003*; *Edelman et al., 2005*; *Goldberg et al., 1996*). This mode of regulation has been well-studied for CaMKI, whose C-terminal tail makes direct contact with the kinase domain, directly inhibiting its enzymatic activity (*Goldberg et al., 1996*). However, it is unclear if, or how, the C-terminal tail of DCLK1 regulates its kinase domain. In addition, the physiological significance of DCLK1 kinase activity is unknown, even though it is a target for the development of kinase inhibitors due to its prominent role in cancer (*Westphalen et al., 2017*; *Weygant et al., 2014*; *Ferguson et al., 2020*). Additional information on the functional role of the DCLK1 kinase domain and how it is controlled would therefore be valuable for understanding how drugs can effectively target DCLK1 for therapeutic purposes.

Here we present a detailed examination of the microtubule-binding properties of DCLK1 and how they are regulated by its kinase activity. We find that DCLK1 autophosphorylates one key residue (T688) within its C-terminal tail via an intramolecular mechanism to strongly modulate its microtubule-binding affinity. Removal of the C-terminal tail or mutation of T688 results in an increase in phosphorylation of residues within both the DC1 and the DC2 domains, which in turn decreases microtubule binding. Furthermore, we observed that mutating four key phosphosites within DC1 of DCLK1 rescues microtubule binding in the construct lacking the C-terminal tail. Overall, our data led to a model in which DCLK1 autophosphorylates its C-terminal tail to modulate the activity of its own kinase domain and, subsequently, the level of phosphorylation within its microtubule-binding domains. To our knowledge, this is the first example of a self-regulatory MAP that can tune its microtubule-binding properties based on autophosphorylation state. Our results uncover a novel intramolecular regulation of microtubule binding within a prominent family of MAPs and may have implications for DCLK1's known roles in tumor development and cancer progression.

## Results

Previous results have suggested that phosphorylation of DCLK1 occurs in part via autophosphorylation (*Patel et al., 2016*; *Shang et al., 2003*). To determine if DCLK1 phosphorylation is mediated by an inter- or intramolecular mechanism, we utilized an established kinase-dead mutant of DCLK1 (D511N) (*Patel et al., 2016*; *Patel et al., 2021*) and an active wild-type (WT) DCLK1 enzyme, both purified from bacteria (*Figure 1—figure supplement 1A* and *Figure 1—figure supplement 2A*). We did not observe trans-phosphorylation of DCLK1-D511N upon incubation with DCLK1-WT, although DCLK1-WT efficiently autophosphorylated itself in this assay (*Figure 1—figure supplement 2B*). Thus, under the conditions in our experiments, DCLK1 phosphorylation occurs via an intramolecular mechanism.

Removal of the C-terminal region of DCLK1 that follows the kinase domain results in an increase in kinase activity (*Shang et al., 2003*). How this region regulates enzymatic activity and autophosphorylation of DCLK1 and how phosphorylation of the molecule affects its microtubule-binding properties are open questions. We first compared the mobility of full-length mouse DCLK1-WT (aa 1–740) and a truncated DCLK1 lacking the C-terminal tail (ΔC: aa 1–648) to full-length kinase-dead DCLK1-D511N on a Phos-tag gel, which enhances the separation of differentially phosphorylated proteins (*Figure 1B* and *Figure 1—figure supplement 1A-B*; *Kinoshita et al., 2009*). We found that bacterially expressed DCLK1-WT and DCLK1-ΔC proteins migrated more slowly into the Phos-tag

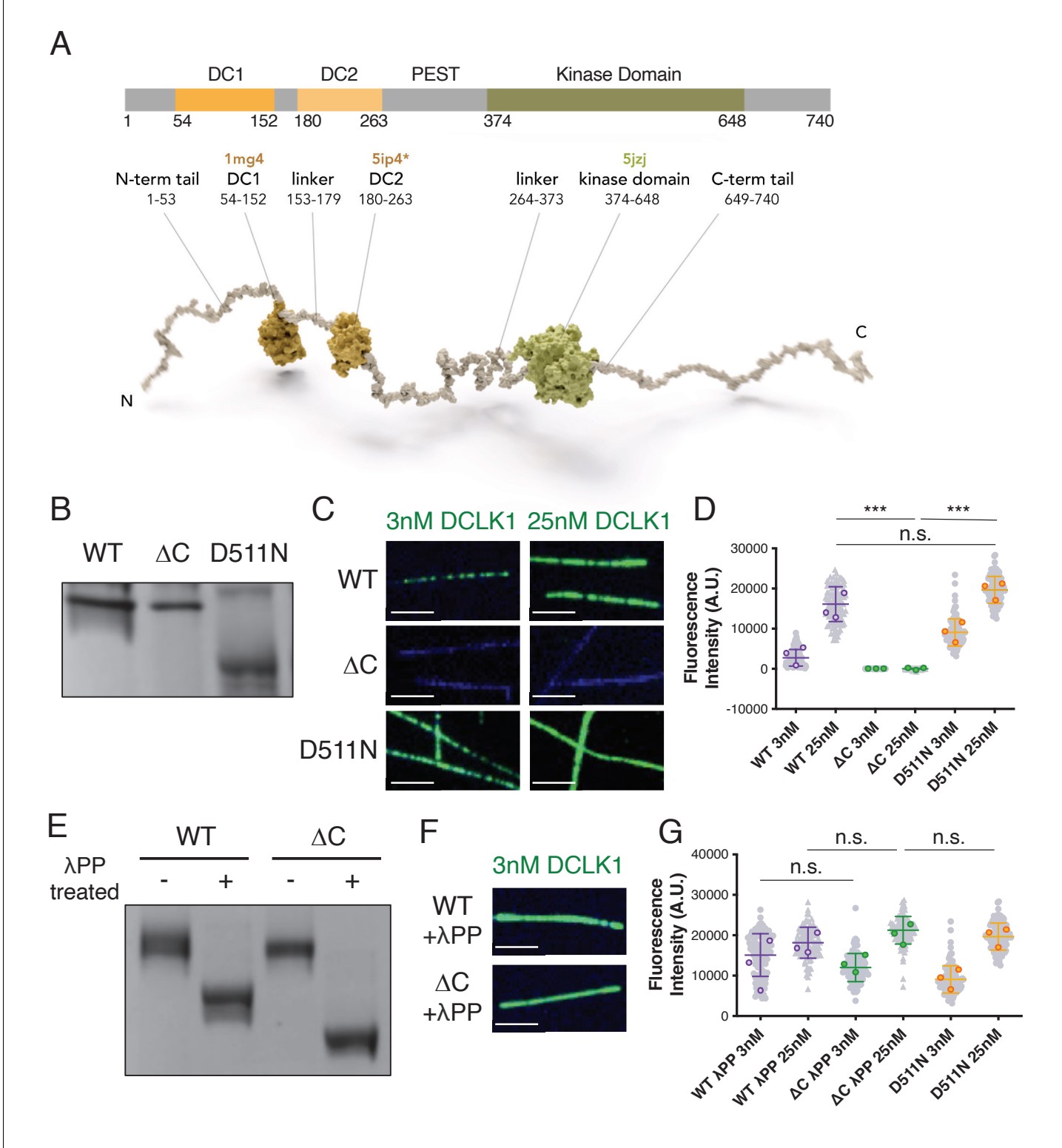

**Figure 1.** The C-terminal domain of DCLK1 regulates autophosphorylation and microtubule binding. (**A**) Diagram of domains and motifs of human doublecortin-like kinase 1 (DCLK1) (UniProt O15075) that are conserved in the mouse DCLK1 used in this study. DC1, N-terminal doublecortin-like (DCX) domain; DC2, C-terminal DCX domain; kinase domain. Motifs enriched in PEST (proline/P, glutamic acid/E, serine/S, threonine/T) and DEND (aspartic acid/D, glutamic acid/E, asparagine/N, aspartic acid/D) based on *Burgess and Reiner, 2001* and *Nagamine et al., 2011*. Below: model of human DCLK1. DC1 domain (1mg4; *Kim et al., 2003a*), DC2 domain modeled by homology to DCX-DC2 (5ip4; *Burger et al., 2016*), kinase domain

*Figure 1 continued on next page*

Figure 1 continued

(5jzj; *Patel et al., 2016*). DCLK1 is shown as a full-length pseudo-model, with projection domains/tails (and domain linkers) modeled as unfolded to visualize the length and convey the intrinsic disorder predicted for those regions. The mouse DCLK1 (1–740) used in this paper has the same amino acid boundaries as that of humans. (B) Coomassie blue-stained sodium dodecyl sulfate-polyacrylamide gel electrophoresis (SDS-PAGE) Phos-tag gel of purified wild-type (WT), ΔC, and kinase-dead (D511N) DCLK1 proteins separated by phosphorylation level. Representative gel from n = 3 independent experiments. (C) Total internal reflection fluorescence microscopy (TIRF-M) images of 3 nM and 25 nM sfGFP-DCLK1 WT, ΔC, and D511N (green), expressed in bacteria under standard conditions, binding to taxol-stabilized microtubules (blue). Scale bars: 2.5 µm. (D) Quantification of microtubule-bound sfGFP-DCLK1 fluorescence intensity. Means ± sd: 2748.9 ± 2073.6 for 3 nM WT, 16119.6 ± 4324.3 for 25 nM WT, 1.2 ± 31.6 for 3 nM ΔC, 3.8 ± 101.9 for 25 nM ΔC, 9072.4 ± 3380.1 for 3 nM D511N, and 19666.6 ± 3345.3 for 25 nM D511N (n>100 microtubules from n = 3 independent trials for each concentration of each protein). Gray dots indicate individual microtubule intensities, while colored dots represent the averages from each trial. ***p<0.0001 and p = 0.3240 for 25 nM WT vs 25 nM D511N, calculated using Student's t-test. p-values were calculated using n = 3. (E) Coomassie blue-stained SDS-PAGE Phos-tag gel of purified DCLK1-WT and -ΔC incubated with lambda phosphatase (λPP) or incubated in buffer alone for 1 hr at 30°C. Representative gel from n = 3 independent experiments. (F) TIRF-M images of 3 nM sfGFP-DCLK1 WT and ΔC (green) after treatment with λPP, binding to taxol-stabilized microtubules (blue). Scale bars: 2.5 µm. (G) Quantification of microtubule-bound sfGFP-DCLK1 fluorescence intensity. Means ± sd: 15090.7 ± 5285.6 for 3 nM WT + λPP, 18155.3 ± 3833.5 for 25 nM WT + λPP, 12004.2 ± 3490.3 for 3 nM ΔC + λPP, and 21240.6 ± 3413.5 for 25 nM ΔC + λPP (n>100 microtubules from n = 3 independent trials for each protein concentration; gray dots indicate individual microtubule intensities, while colored dots represent the averages from each trial.). D511N data are reproduced from (D) for comparison. p = 0.3566 for 25 nM WT + λPP vs 25 nM ΔC + λPP, p = 0.6341 for 25 nM WT + λPP vs 25 nM D511N, p = 0.5989 for 25 nM ΔC + λPP vs 25 nM D511N, and p = 0.4462 for 3 nM WT + λPP vs 3 nM ΔC + λPP, calculated using Student's t-test. p-values were calculated using n = 3. For all experiments, at least two separate protein purifications were used.

The online version of this article includes the following source data and figure supplement(s) for figure 1:

**Source data 1.** Uncropped gels for the associated panels in *Figure 1*.
**Source data 2.** Uncropped gels.
**Figure supplement 1.** Diagram of full-length DCLK1 used in this study and gels for purified recombinant proteins used in this study.
**Figure supplement 1—source data 1.** Uncropped gels for the associated panels in *Figure 1—figure supplement 1*.
**Figure supplement 1—source data 2.** Uncropped gels.
**Figure supplement 2.** DCLK1 autophosphorylates via an intramolecular mechanism.
**Figure supplement 2—source data 1.** Uncropped blots (A, B), gels (C, E) for the associated panels in *Figure 1—figure supplement 2*.
**Figure supplement 2—source data 2.** Uncropped gels.

gel, indicative of higher levels of phosphorylation, compared to the non-phosphorylated DCLK1-D511N (*Figure 1B*). Using total internal reflection fluorescence microscopy (TIRF-M), we imaged sfGFP-tagged WT, ΔC, and D511N proteins binding to taxol-stabilized microtubules (*Figure 1C*) at concentrations differing by eightfold. Strikingly, DCLK1-ΔC did not bind to microtubules at either concentration tested, in stark contrast to DCLK1-WT and DCLK1-D511N, which both robustly bound to microtubules (*Figure 1C–D*). Notably, DCLK1-D511N bound microtubules more robustly at lower concentrations than DCLK1-WT. This is consistent with the prior work showing that D511N robustly stimulates tubulin polymerization (*Patel et al., 2016*). These experiments suggest that the C-terminal region regulates DCLK1 autophosphorylation, which in turn directly modulates its microtubule-binding affinity.

To test this possibility, we sought to evaluate the microtubule-binding behaviors of dephosphory-lated DCLK1-WT and DCLK1-ΔC. We incubated the proteins with the $Mn^{2+}$-dependent protein phosphatase, lambda phosphatase (λPP), which strongly dephosphorylated DCLK1-WT and DCLK1-ΔC as evidenced by the marked shifts on a Phos-tag gel without phospho-intermediate bands, but had little effect on the migration of D511N (*Figure 1E* and *Figure 1—figure supplement 2C*). The similar migration of DCLK1-WT and DCLK1-ΔC in the absence of phosphatase, coupled with the relatively larger migration shift of DCLK1-ΔC after treatment, suggests that DCLK1-ΔC is hyperphosphorylated compared to the WT protein, in agreement with previous results (*Shang et al., 2003*). In addition, anion exchange chromatograms also revealed a greater shift in the elution volume between phosphorylated and non-phosphorylated DCLK1-ΔC compared to the shift observed for phosphorylated vs non-phosphorylated WT protein, consistent with DCLK1-ΔC having a higher negative charge due to being hyperphosphorylated (*Figure 1—figure supplement 1C*). Using TIRF-M, we found that λPP-treated DCLK1-WT and DCLK1-ΔC bound to microtubules similarly to DCLK1-D511N (*Figure 1F–G*), similar to prior results showing that phosphatase-treated DCLK1 robustly stimulates tubulin polymerization (*Patel et al., 2016*). This further suggests that autophosphorylation

modulates the microtubule-binding affinity of DCLK1 and that hyperphosphorylation of DCLK1-ΔC largely abolishes microtubule binding.

The high phosphorylation levels observed for both DCLK1-WT and DCLK1-ΔC suggested that these proteins phosphorylate themselves during bacterial expression. To determine the contributions of the C-terminal tail to DCLK1 function, we used a previously defined strategy to control the levels of autophosphorylation during expression (*Patel et al., 2016*; *Patel et al., 2021*). We co-expressed DCLK1-WT and DCLK1-ΔC with λPP in bacteria, followed by the subsequent removal of λPP from the DCLK1 preps via affinity and ion exchange chromatography. We compared DCLK1 proteins prepared in the absence or presence of λPP on a Phos-tag gel and observed that λPP co-expression substantially reduced phosphorylation levels of both DCLK1-WT and DCLK1-ΔC (*Figure 2A*). For all subsequent experiments, all DCLK1 protein variants were co-expressed with λPP. Upon incubation of dephosphorylated DCLK1 proteins with adenosine triphosphate (ATP), both DCLK1-WT and DCLK1-ΔC exhibited an increase in phosphorylation, but DCLK1-ΔC appeared to be entirely phosphorylated by 30 min, whereas DCLK1-WT displayed a number of phosphorylated intermediates even at 60 min (*Figure 2A–B*; 93.7% of DCLK1-ΔC protein shifts to the uppermost band after a 30-min incubation with ATP compared with 41.8% of DCLK1-WT protein after a 60-min incubation with ATP). Using TIRF-M, we determined the microtubule-binding affinities for DCLK1-WT and DCLK1-ΔC in the absence and presence of ATP (*Figure 2C–E*). We found that, in the absence of ATP, both proteins exhibited relatively similar microtubule-binding affinities (*Figure 2C–E*). After a 30-min incubation with ATP, the microtubule-binding affinity of DCLK1-WT moderately weakened, as evidenced by an ~2.5-fold increase in $K_D$ (*Figure 2D*). However, incubation with ATP resulted in a dramatic approximately forty-onefold decrease in microtubule affinity of DCLK1-ΔC (*Figure 2E*). Interestingly, in the presence of ATP, DCLK1-ΔC was still present at regions of microtubule curvature, consistent with prior results that doublecortin proteins have a higher affinity for these regions in vitro (*Figure 1—figure supplement 2D*; *Bechstedt et al., 2014*). We also analyzed the binding behaviors of WT and ΔC on non-stabilized guanosine diphosphate (GDP) microtubule lattices grown from GMPCPP seeds and observed a similar decrease in bound DCLK1-ΔC in the presence of ATP (*Figure 2F–G*). Finally, we performed a microtubule co-sedimentation assay with WT and ΔC and found that while similar amounts of protein pelleted with microtubules in the absence of ATP, significantly less ΔC co-pelleted with microtubules in the presence of ATP (*Figure 1—figure supplement 2E*). These results indicate that the loss of its regulatory C-terminal tail results in aberrant DCLK1 hyperphosphorylation, leading to a dramatic loss of microtubule-binding affinity. Thus, the kinase activity of DCLK1 directly controls its association with microtubules via intramolecular phosphorylation, which in turn is regulated by the C-terminus of the protein.

In order to determine how phosphorylation regulates the microtubule-binding affinity of DCLK1-ΔC, we performed liquid chromatography with tandem mass spectrometry (LC-MS/MS) of phosphorylated DCLK1-WT and DCLK1-ΔC proteins. For each phosphorylated residue identified, we counted the total number of DCLK1-WT and DCLK1-ΔC peptides containing the phosphorylated residue, and then calculated the percent of those peptides whose spectra revealed phosphorylation at that residue. Within the microtubule-binding region of DCLK1 (aa 44–263), we found that nine sites were more frequently phosphorylated in DCLK1-WT samples and 17 sites were more frequently phosphorylated in DCLK1-ΔC samples (*Figure 3A–B*). Of the 17 phosphorylation sites in DCLK1-ΔC, five either directly contact tubulin or are adjacent to residues that directly contact tubulin within the lattice (*Manka and Moores, 2019*). The architecture of DCLK1 suggests that it likely has the flexibility to autophosphorylate its N-terminal half due to an intrinsically disordered region between the DC2 domain and the kinase domain (*Figure 3C*; aa 263–374). These results indicate that the loss of microtubule binding of DCLK1-ΔC is due to an increase in phosphorylation at multiple sites, as opposed to a single site whose phosphorylation status dictates microtubule binding.

The decrease in microtubule-binding affinity we observed for the DCLK1-ΔC construct in the presence of ATP could be due to a general increase in phosphorylation throughout the entire microtubule-binding region (aa 46–263) or due to phosphorylation at specific sites within the microtubule-binding DC domains. In order to determine the relative contributions of phosphorylation within the DC1 and DC2 domains to the decrease in the microtubule-binding affinity of DCLK1-ΔC, we mutated four conserved residues that showed the highest increase in phosphorylation within the DC1 ($C_{4A\text{-}DC1}$: S77, S83, S96, T143) or the DC2 domain ($C_{4A\text{-}DC2}$: T189, S193, T218, S228) (*Figure 4* and *Figure 1—figure supplement 1A-B*, *Figure 3—figure supplement 1A-B*; *Manka and Moores, 2019*).

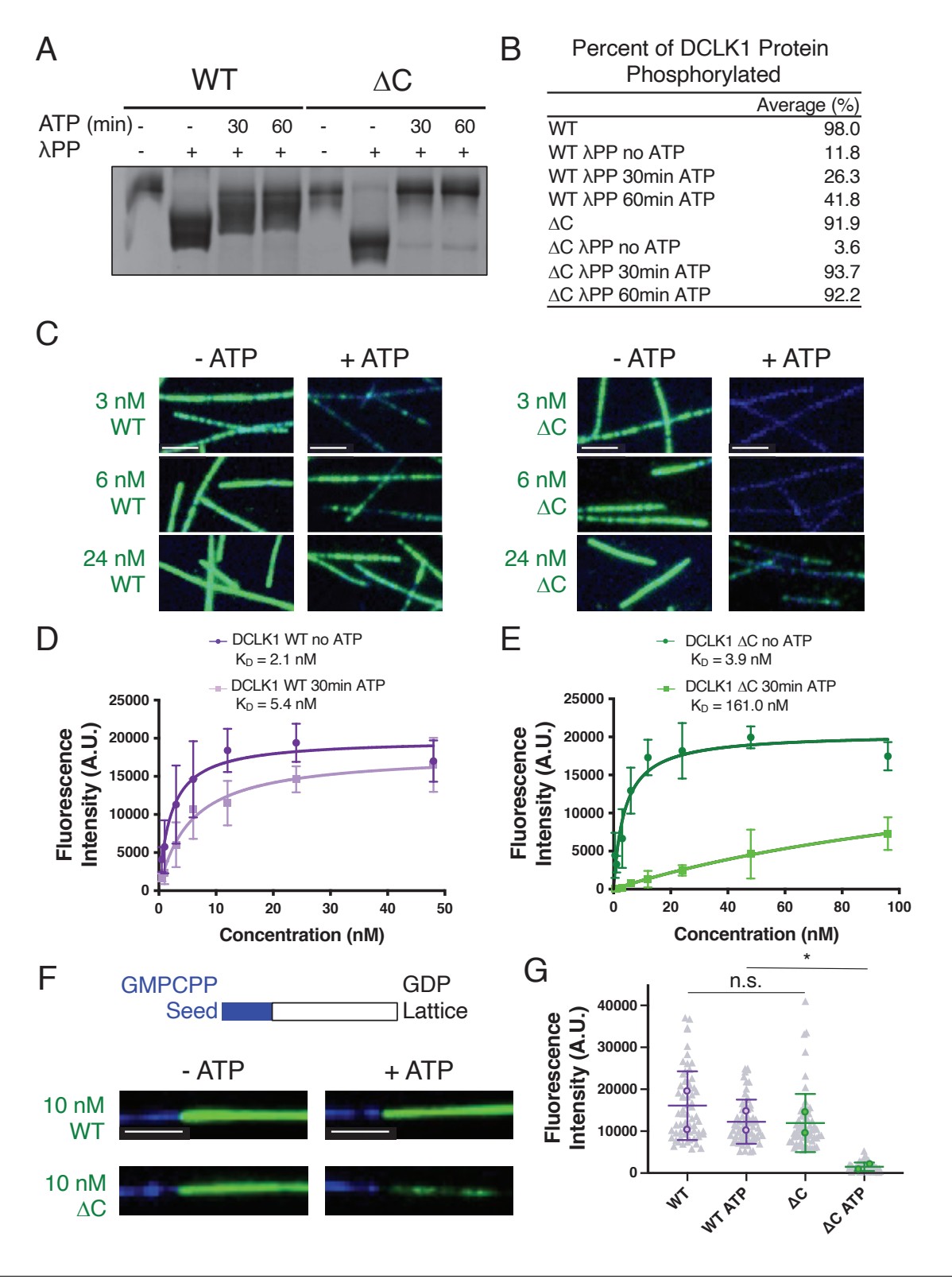

**Figure 2.** Hyperphosphorylation of DCLK1-ΔC prohibits microtubule binding. (**A**) Coomassie blue-stained sodium dodecyl sulphate–polyacrylamide gel electrophoresis (SDS-PAGE) Phos-tag gel of purified doublecortin-like kinase 1 wild-type (DCLK1-WT) and -ΔC proteins separated by phosphorylation level. The first and fifth lanes contain DCLK1-WT and -ΔC expressed in bacteria under standard conditions. All other lanes contain DCLK1-WT and -ΔC that were co-expressed with lambda phosphatase (λPP), which was subsequently separated from DCLK1. Incubation of λPP-treated DCLK1-WT and -ΔC

*Figure 2 continued on next page*

*Figure 2 continued*

with 2 mM adenosine triphosphate (ATP) at the indicated times reveals a band shift, indicative of an increase in phosphorylation. (**B**) Quantification of the average percent of total DCLK1 protein that is phosphorylated in each condition. Averages are derived from n = 3 independent experiments. (**C**) T otal internal reflection fluorescence microscopy (TIRF-M) images of sfGFP-DCLK1-WT and -ΔC (co-expressed in bacteria with λPP) at indicated concentrations (green) binding to taxol-stabilized microtubules (blue) after a 30-min incubation in the absence or presence of 2 mM ATP. Scale bars: 2.5 μm. (**D**) Quantification of microtubule-bound sfGFP-DCLK1-WT fluorescence intensity plotted against concentration after a 30-min incubation in the absence or presence of ATP (WT without ATP, $K_D$ = 2.1 nM, and WT with ATP, $K_D$ = 5.4 nM, derived from at least n = 3 independent trials per condition). (**E**) Quantification of microtubule-bound sfGFP-DCLK1-ΔC fluorescence intensity plotted against concentration after a 30-min incubation in the absence or presence of ATP (ΔC without ATP, $K_D$ = 3.9 nM, and ΔC with ATP, $K_D$ = 161.0 nM, derived from n = 3 independent trials). (**F**) TIRF-M images of 10 nM sfGFP-DCLK1-WT or -ΔC (green, co-expressed in bacteria with λPP) binding to non-stabilized GDP microtubules grown from GMPCPP seeds (blue) after a 30-min incubation in the absence or presence of 2 mM ATP. Scale bars: 2.5 μm. (**G**) Quantification of microtubule-bound sfGFP-DCLK1 fluorescence intensity. Means ± sd: 15004.4 ± 6503.8 for WT, 12535.9 ± 3247.5 for WT + ATP, 12111.3 ± 3534.0 for ΔC, and 1579.3 ± 866.5 for ΔC + ATP (n>60 microtubules from n = 2 independent trials for each condition; gray dots indicate individual microtubule intensities, while colored dots represent the averages from each trial; p = 0.6360 for WT vs ΔC and p = 0.0440 for WT + ATP vs ΔC + ATP, calculated using Student's t-test; p-values were calculated using n = 2). For all experiments, at least two separate protein purifications were used.

The online version of this article includes the following source data for figure 2:

**Source data 1.** Uncropped gel for the associated panel in *Figure 2*.
**Source data 2.** Uncropped gels.

We reasoned that if phosphorylation of these residues is responsible for the decreased microtubule-binding affinity of DCLK1-ΔC, then mutating these residues to alanines, which cannot be phosphorylated, should rescue the microtubule-binding defect of this construct in the presence of ATP. If, however, phosphorylation of these residues is not responsible for the decreased microtubule-binding affinity, then there should be no difference in binding between the DCLK1-ΔC and ΔC$_{4A-DC1}$ or C$_{4A-DC2}$ regardless of the presence of ATP.

Using TIRF-M, we imaged DCLK1-ΔC, ΔC$_{4A-DC1}$, and C$_{4A-DC2}$ binding to taxol-stabilized microtubules in the presence or absence of ATP (*Figure 4B–D*). In the absence of ATP, all three DCLK1 proteins bound robustly to microtubules at both 5 nM and 20 nM. For all proteins at 5 nM, in the presence of ATP, there was substantially less DCLK1 on the microtubule (*Figure 4B–C*), similar to our previous results (*Figure 2*); however, there was a small, but significant increase in the amount of ΔC$_{4A-DC1}$ bound to the microtubule (*Figure 4B–C*). At 20 nM, in the presence of ATP, we observed significantly more ΔC$_{4A-DC1}$ and ΔC$_{4A-DC2}$ on the microtubule compared with ΔC, but this result was far more striking for ΔC$_{4A-DC1}$, which exhibited an approximately eightfold higher fluorescence intensity on the microtubule than ΔC (*Figure 4B,D*). These results indicate that mutating the residues in DC1 that are abnormally phosphorylated in the ΔC construct rescued microtubule binding by preventing phosphorylation within this domain. Mutating the phosphosites within DC2 also rescued microtubule binding, but to a lesser extent than the DC1 mutations. Therefore, aberrant phosphorylation of these residues in DCLK1-ΔC could indeed be responsible for the dramatic attenuation of microtubule binding.

We next wanted to determine the mechanism by which the C-terminal region of DCLK1 prevents hyperphosphorylation of the DC domains. We examined autophosphorylated DCLK1-WT by LC-MS/MS and identified two threonine residues in the C-terminal region (T687 and T688) that were consistently phosphorylated (*Figure 3—figure supplement 1C*). In order to understand how the C-terminal tail contributes to autophosphorylation, we mutated T687 and T688 to alanines individually (*Figure 5A*). For all of the experiments with these mutants, we co-expressed DCLK1 proteins with λPP to obtain a dephosphorylated protein preparation. We first evaluated the ability of these mutants to autophosphorylate using the Phos-tag gel system (*Figure 5B–C*). While DCLK1-WT and T687A exhibited a moderate increase in phosphorylation after a 30-min incubation with ATP, T688A appeared to be entirely phosphorylated at this same time point (*Figure 5B–C*; 13.2, 62.7, and 96.0% of protein shifts to the most phosphorylated band after a 30-min incubation with ATP for WT, T687A, and T688A, respectively). Therefore, phosphorylation of T688 within the C-terminal domain may be critical for the regulation of the kinase activity of DCLK1. To further elucidate the consequences of abolishing these phosphorylation sites, we used TIRF-M to determine microtubule-binding

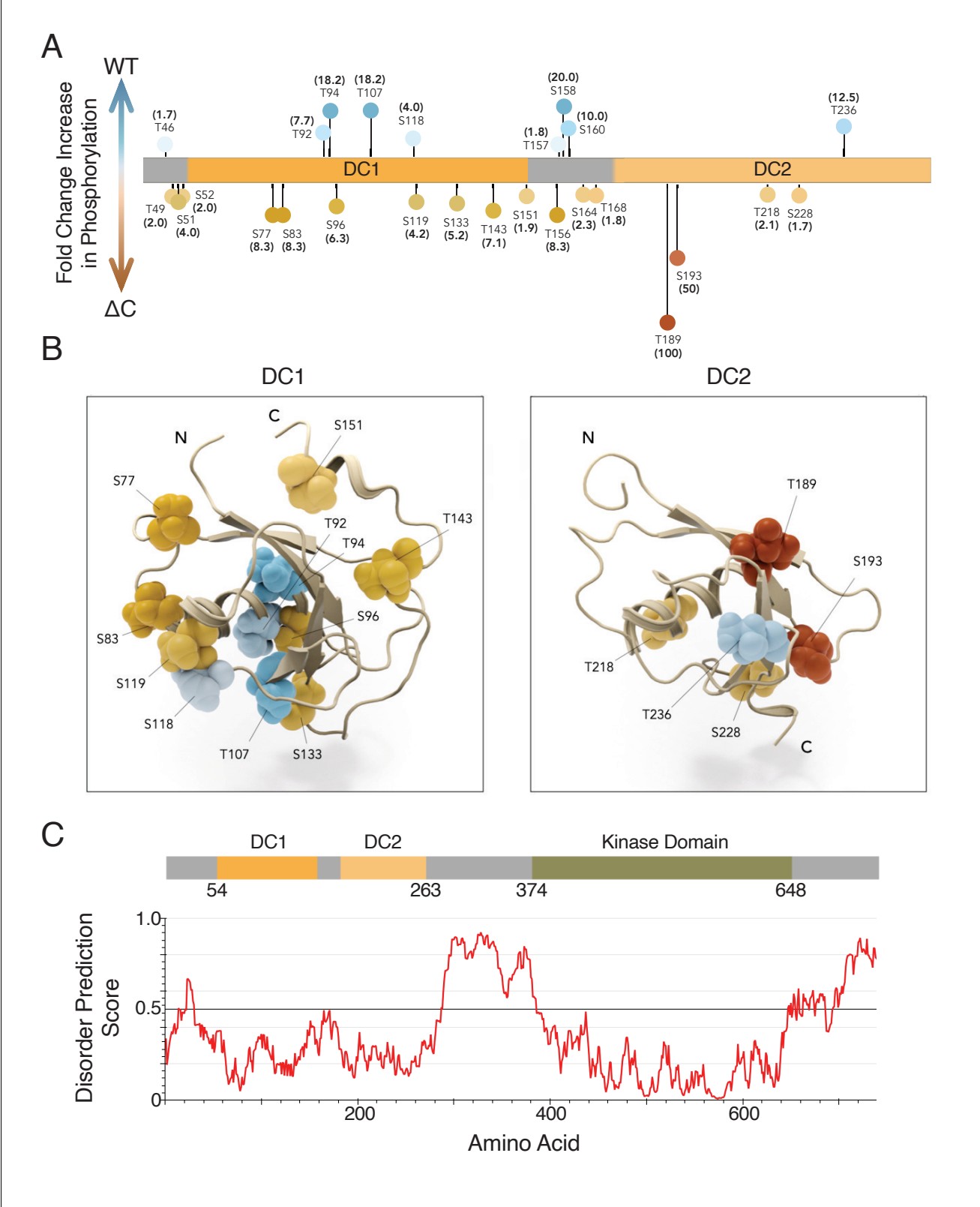

**Figure 3.** DCLK1-ΔC aberrantly autophosphorylates at multiple sites within the microtubule-binding region. (**A, B**) Visualization of changes in phosphorylation due to deletion of C-terminal domain. Experiment compared peptides from wild-type (WT) and ΔC constructs: data are expressed as fold-change increases in phosphorylation in one construct over the other based on the percent of total peptides that exhibited phosphorylation at a particular site (n = 751 and 637 total peptides analyzed for WT and ΔC, respectively, from n = 3 independent experiments). Darker colors indicate a

*Figure 3 continued on next page*

*Figure 3 continued*

higher fold change in phosphorylation; that is darker blue indicates that a site is more commonly phosphorylated in the WT construct, while darker orange indicates a site is more commonly phosphorylated in the ΔC construct. (**A**) Lollipop plot summarizes changes in phosphorylation (≥1.5 fold change) mapped onto a diagram of the mouse doublecortin-like kinase 1 (DCLK1) used in this study, but all listed residues are conserved in human DCLK1. (**B**) DC1 domain (1mg4; *Kim et al., 2003a*) and DC2 domain modeled by homology to DCX-DC2 (5ip4; *Burger et al., 2016*). Domain structures are aligned and shown as ribbon representations with labeled S/T residues visualized as CPK/balls. Level of saturation in color indicates fold change in phosphorylation of those residues: increase in WT (blue colors) and increase in ΔC (orange/red colors). (**C**) Architecture of DCLK1 protein with the per-residue IUPRED2A (*Mészáros et al., 2018*) disorder prediction score shown in the corresponding plot with a cutoff value of 0.5 indicated by the dashed line. Residues scored above this value are predicted to be disordered.

The online version of this article includes the following figure supplement(s) for figure 3:

**Figure supplement 1.** Dissection of the phosphorylated residues within DCLK1.

affinities for the DCLK1 phosphomutants in the presence or absence of ATP (*Figure 5D–F*). In the absence of ATP, all DCLK1 proteins exhibited relatively similar microtubule-binding affinities based on the dissociation constants derived from fluorescent saturation curves (*Figure 5D–F*). After a 30 min incubation with ATP, the microtubule-binding affinities of DCLK1-WT and T687A were comparable, whereas T688A displayed a dramatic reduction in microtubule binding with an approximately fortyfold increase in $K_D$ (*Figure 5F*). We also performed a microtubule co-sedimentation assay with WT and T688A and found that while similar amounts of protein pelleted with microtubules in the absence of ATP, significantly less T688A co-pelleted with microtubules in the presence of ATP (*Figure 5G*). These results indicate that DCLK1 likely autophosphorylates residues within its C-terminal region in order to control aberrant hyperphosphorylation within its microtubule-binding domain.

There could be consequences of autophosphorylation for DCLK1 function other than microtubule binding. We therefore analyzed the effect of autophosphorylation on DCLK1 conformation and on DCLK1 sensitivity to calpain cleavage. First, we fractionated the following proteins by sucrose density centrifugation: normally expressed DCLK1-WT, DCLK1-WT co-expressed with λPP, normally expressed DCLK1-T688A, DCLK1-T688A co-expressed with λPP, and DCLK1-D511N (*Figure 5—figure supplement 1A–B*). We observed a similar profile for all proteins on a 3–9% step gradient (*Figure 5—figure supplement 1A–B*), indicating these proteins adopt similar gross conformations regardless of phosphorylation state. DCLK1 contains two PEST (proline/P, glutamic acid/E, serine/S, threonine/T) domains that are targeted by calpain for proteolytic cleavage (*Figure 1A*; *Patel et al., 2016*; *Burgess and Reiner, 2001*). Upon incubation of calpain with λPP-co-expressed DCLK1-WT or DCLK1-T688A that had been first incubated in the absence or presence of ATP, we detected similar cleavage products of DCLK1 for both proteins under both conditions (*Figure 5—figure supplement 1C*). Therefore, autophosphorylation does not appear to affect cleavage of purified DCLK1-WT by calpain under our conditions. These results support a regulatory role for autophosphorylation in dictating the microtubule-binding affinity of DCLK1, without affecting the overall, gross conformation of the molecule or its sensitivity to calpain cleavage.

## Discussion

Overall, our study elucidates a mechanism by which DCLK1 modulates its kinase activity to tune its microtubule-binding affinity. We have found that DCLK1 autophosphorylates within its C-terminal tail to prevent aberrant hyperphosphorylation within its microtubule-binding domain. To our knowledge, this is the first example of a MAP whose binding is controlled by autophosphorylation. Based on the relevancy of DCLK1 to the progression of a variety of cancer types, understanding DCLK1 autoregulation is critical in determining its biological function in healthy versus disease states.

Autophosphorylation control of microtubule-binding affinity is likely to be controlled by cellular context. DCX is tightly regulated at the transcriptional level to ensure proper temporal expression during neuronal development (*Francis et al., 1999*; *Edelman et al., 2005*; *des Portes et al., 1998*; *Gleeson et al., 1999*). In contrast, members of the DCLK family are expressed during embryonic, post-embryonic, and adult periods, and are also expressed in a range of tissues outside the brain

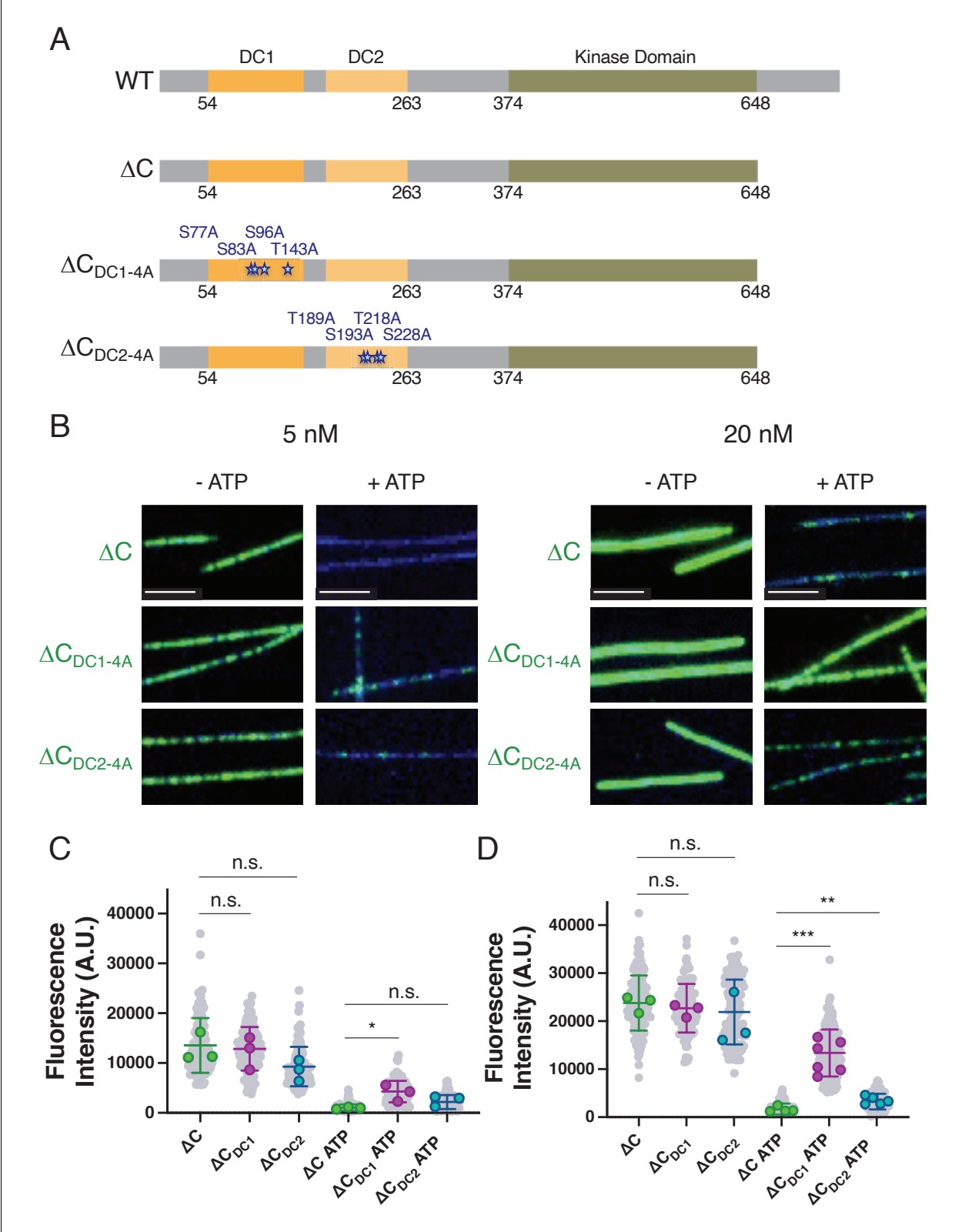

**Figure 4.** Phosphonull mutations within DC1 restore microtubule binding of ΔC. (**A**) Diagrams depicting the domains, amino acid boundaries, and mutations relevant to the doublecortin-like kinase 1 (DCLK1) constructs used. $\Delta C_{DC1-4A}$ indicates the four residues in DC1 that were mutated to alanines S77, S83, S96, and T143. $\Delta C_{DC2-4A}$ indicates the four residues in DC2 that were mutated to alanines T189, S193, T218, and S228. (**B**) Total internal reflection fluorescence microscopy (TIRF-M) images of sfGFP-DCLK1 ΔC, $\Delta C_{DC1-4A}$, and $\Delta C_{DC2-4A}$, co-expressed in bacteria with lambda

*Figure 4 continued on next page*

*Figure 4 continued*

phosphatase (λPP), at indicated concentrations binding to taxol-stabilized microtubules (blue) in the absence or presence of adenosine triphosphate (ATP). Scale bars: 2.5 µm. (C) Quantification of microtubule-bound 5 nM sfGFP-DCLK1 fluorescence intensity. For 5 nM concentrations in the absence of ATP, means ± sd: 12883.5 ± 2881.6 for ΔC, 12245.5 ± 3283.0 for ΔC$_{DC1-4A}$, 8552.7 ± 2097.3 for ΔC$_{DC2-4A}$ (n>100 microtubules per condition from n = 3 independent trials; gray dots indicate individual microtubule intensities, while colored dots represent the averages from each trial; p = 0.8128 for ΔC vs ΔC$_{DC1-4A}$ and p = 0.1031 for ΔC vs ΔC$_{DC2-4A}$ calculated using Student's t-test; p-values were calculated using n = 3). For 5 nM concentrations in the presence of ATP, means ± sd: 987.6 ± 202.5 for ΔC + ATP, 4042.6 ± 1624.6 for ΔC$_{DC1-4A}$ + ATP, 2482.0 ± 1058.3 for ΔC$_{DC2-4A}$ + ATP (n>100 microtubules from n = 3 independent trials; gray dots indicate individual microtubule intensities, while colored dots represent the averages from each trial; p = 0.0319 for ΔC vs ΔC$_{DC1-4A}$ and p = 0.0742 for ΔC vs ΔC$_{DC2-4A}$ calculated using Student's t-test; p-values were calculated using n = 3). (D) Quantification of microtubule-bound 20 nM sfGFP-DCLK1 fluorescence intensity. For 20 nM concentrations in the absence of ATP, means ± sd: 23634.3 ± 1725.1 for ΔC, 22277.3 ± 1334.9 for ΔC$_{DC1-4A}$, 19912.2 ± 5408.6 for ΔC$_{DC2-4A}$ (n>100 microtubules per condition from n = 3 independent trials; gray dots indicate individual microtubule intensities, while colored dots represent the averages from each trial; p = 0.3419 for ΔC vs ΔC$_{DC1-4A}$ and p = 0.3196 for ΔC vs ΔC$_{DC2-4A}$ calculated using Student's t-test; p-values were calculated using n = 3). For 20 nM concentrations in the presence of ATP, means ± sd: 1579.8 ± 585.1 for ΔC + ATP, 12556.9 ± 3419.9 for ΔC$_{DC1-4A}$ + ATP, 3503.4 ± 826.7 for ΔC$_{DC2-4A}$ + ATP (n>100 microtubules from n = 4, 6, and 5 independent trials for ΔC, ΔC$_{DC1-4A}$, and ΔC$_{DC2-4A}$, respectively; gray dots indicate individual microtubule intensities, while colored dots represent the averages from each trial; p = 0.0002 for ΔC vs ΔC$_{DC1-4A}$ and p = 0.0058 for ΔC vs ΔC$_{DC2-4A}$ calculated using Student's t-test; p-values were calculated using n = number of independent trials as stated above). For all experiments, at least two separate protein purifications were used.

(*Reiner et al., 2006*; *Edelman et al., 2005*). This lack of temporal restriction may, therefore, necessitate autoregulation, as well as other modes of control, for the DCLK family in a range of cellular activities. It is tempting to speculate that DCLK1 autophosphorylation acts as a temporal switch to control microtubule-binding affinity under specific situations, and this period of activity could be extended or shortened by modulation by phosphatases, other kinases, and proteases.

Although DCX and DCLKs have been found to nucleate, tip-track, and bundle microtubules, the physiological functions of these proteins in regulating microtubule growth and organization remain elusive (*Moores et al., 2004*; *Moores et al., 2006*; *Bechstedt and Brouhard, 2012*; *Bechstedt et al., 2014*; *Ettinger et al., 2016*). Understanding the conserved mechanisms by which these paralogs bind to microtubules is essential in ascribing molecular functions to DCX family members, most of which are implicated in disease (*Reiner et al., 2006*). Prior work has demonstrated that isolated DC domains cannot stimulate microtubule polymerization or effectively bind microtubules, indicating both domains are required in tandem (*Sapir et al., 2000*; *Kim et al., 2003a*; *Taylor et al., 2000*; *Kim et al., 2003b*). The contributions of the individual DC domains are still controversial. Initial cryo-electron microscopy (EM) structures of DCX on microtubules revealed only a single bound DC domain, which was hypothesized to be DC1 (*Moores et al., 2006*; *Fourniol et al., 2013*). However, a subsequent study showed that specifically blocking DC2, but not DC1, prevented DCX from interacting with microtubules, suggesting DC2 is critical for lattice binding (*Burger et al., 2016*). Recent cryo-EM data of DC1 and DC2 bound to microtubule lattices in different nucleotide states unveiled that DC2 binds to the guanosine triphosphate (GTP) microtubule lattice, while DC1 prefers the GDP microtubule lattice (*Manka and Moores, 2019*). This model is consistent with observations that DCX tracks the growing plus-end of the microtubule, which is specifically disrupted by missense mutations in DC2 (*Bechstedt et al., 2014*). Our data indicate that in the absence of the C-terminal tail (or mutation of T688), DCLK1 aberrantly phosphorylates itself within both the DC1 and DC2 domains, but it is primarily the phosphorylation within the DC1 that reduces the microtubule-binding affinity of the molecule. Mutating four phosphosites within DC1 restored microtubule binding for DCLK1-ΔC in the presence of ATP; however, we also observed a small, but significant, rescue of microtubule binding upon mutating four phosphosites within DC2, indicating the importance of this domain (*Figure 4*). It will be imperative in the future to determine the individual and tandem roles of the DC domains across the DCX superfamily in a range of microtubule processes.

We have found that phosphorylation at different sites within DCLK1 has distinct effects on the molecule. Phosphorylation within the C-terminal region is essential to restrict the kinase activity of DCLK1, preventing hyperphosphorylation within the DC2 domain and eradication of microtubule binding. This mechanism could prove important in directing the kinase activity of DCLK1 to orthogonal molecular substrates (*Koizumi et al., 2017*) instead of itself. Alternatively, the C-terminal tail of

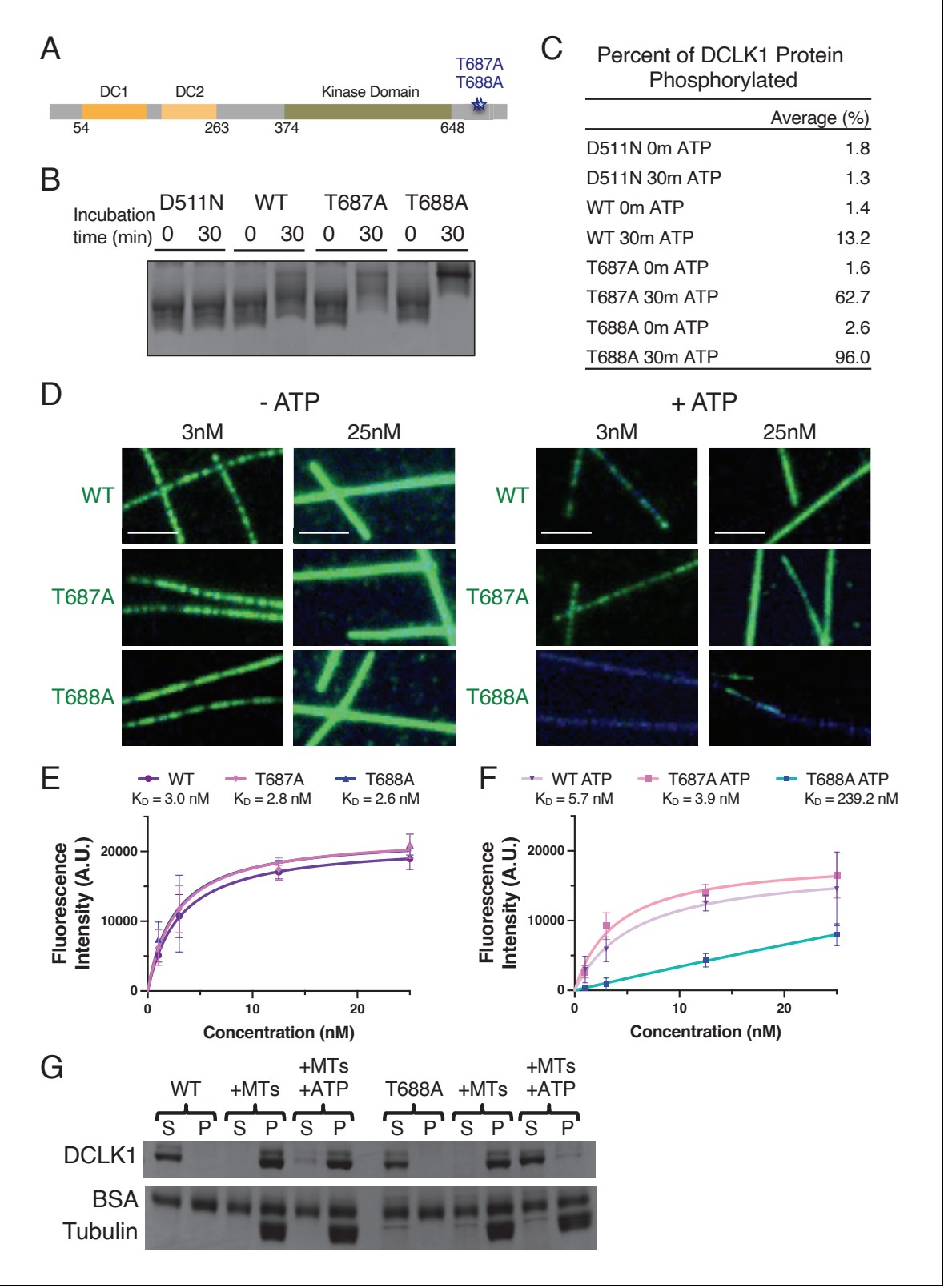

**Figure 5.** Normal autophosphorylation within the C-terminal domain of DCLK1 is necessary to prevent hyperphosphorylation of the rest of the molecule. (**A**) Diagram depicting the domains, amino acid boundaries, and mutations in the C-terminal region relevant to the doublecortin-like kinase 1 (DCLK1) constructs used. (**B**) Coomassie blue-stained sodium dodecyl sulphate–polyacrylamide gel electrophoresis (SDS-PAGE) Phos-tag gel of purified kinase-dead (D511N), WT, T687A, and T688A DCLK1 proteins separated by phosphorylation level. For all experiments, DCLK1 proteins were

*Figure 5 continued on next page*

*Figure 5 continued*

co-expressed with lambda phosphatase (λPP), which was subsequently separated from DCLK1. Incubation of λPP-treated DCLK1 proteins with 2 mM adenosine triphosphate (ATP) at the indicated times reveals a shift in the phosphorylation level to varying degrees. (C) Quantification of the average percent of total DCLK1 protein that is phosphorylated in each condition. Averages were derived from n = 3 independent experiments. (D) Total internal reflection fluorescence microscopy (TIRF-M) images of sfGFP-DCLK1-WT, -T687A, and -T688A (co-expressed in bacteria with λPP) at indicated concentrations (green) binding to taxol-stabilized microtubules (blue) after a 30-min incubation in the absence or presence of 2 mM ATP. Scale bars: 2.5 μm. (E) Quantification of microtubule-bound sfGFP-DCLK1-WT, -T687A, and -T688A fluorescence intensity plotted against concentration after a 30-min incubation in the absence of ATP ($K_D$ = 3.0 nM, 2.8 nM, and 2.6 nM for WT, T687A, and T688A, respectively, from at least n = 3 independent trials per condition). (F) Quantification of microtubule-bound sfGFP-DCLK1-WT, -T687A, and -T688A fluorescence intensity plotted against concentration after a 30-min incubation with ATP ($K_D$ = 5.7 nM, 3.9 nM, and 239.2 nM for WT, T687A, and T688A, respectively, from at least n = 3 independent trials per condition). (G) Coomassie blue-stained SDS-PAGE shows the binding behavior of 500 nM DCLK1-WT or -T688A in the absence or presence of 2 mM ATP in the absence or presence of 2 μM taxol-stabilized microtubules. In the absence of ATP, the percent (means ± sd) of DCLK1 that co-pelleted with microtubules was 99.3 ± 0.5% for WT and 98.9 ± 1.0% for T688A (n = 3 independent experiments; p = 0.5690). In the presence of ATP, the percent (means ± sd) of DCLK1 that co-pelleted with microtubules was 86.2 ± 4.9% for WT and 7.9 ± 3.5% for T688A (n = 3 independent experiments; p<0.0001). For all experiments, at least two separate protein purifications were used.

The online version of this article includes the following source data and figure supplement(s) for figure 5:

**Source data 1.** Uncropped gels for the associated panels in *Figure 5*.
**Source data 2.** Uncropped gels.
**Figure supplement 1.** DCLK1 autophosphorylation does not grossly alter protein conformation or cleavage.
**Figure supplement 1—source data 1.** Uncropped gels for the associated panels in *Figure 5—figure supplement 1*.
**Figure supplement 1—source data 2.** Uncropped gels.

DCLK1 could be cleaved under specific cellular situations to release it from the microtubule (*Burgess and Reiner, 2001*; *Sarkar et al., 2017*). Our assays provide evidence that autophosphorylation does not appear to affect calpain cleavage or the overall gross conformation of DCLK1. However, these results do not rule out the effects of phosphorylation on processing by other proteolytic enzymes or potentially small or dynamic conformational changes that would not be apparent in our assays. Future studies will be necessary to expand upon these results both in vitro and in vivo and determine how DCLK1 function is regulated within the cell.

The region of DCLK1 spanning the kinase domain and C-terminal tail is 46% identical to the comparable region of CaMKI (*Patel et al., 2016*; *Shang et al., 2003*; *Edelman et al., 2005*). The flexible C-terminal tail of CaMKI serves as a regulatory switch; it forms multiple interactions with the kinase domain and keeps it in an inactive conformation (*Goldberg et al., 1996*). Our data implicating the C-terminal tail of DCLK1 in preventing kinase hyperactivity combined with prior structural data propose a similar model for DCLK1 (*Patel et al., 2016*). Furthermore, we have identified a residue (T688) within the DCLK1 C-terminal tail that is critical in modulating kinase activity. Whether the DCLK1 C-terminal tail interacts stably or dynamically with the kinase domain and when DCLK1 may need to switch from a microtubule-bound to -unbound state within the cell are open questions. Overall, the evidence for a conserved mechanism governing kinase activity for CaMKI and DCLK1 and how this could go awry in disease provide exciting new avenues for future exploration.

Finally, this study has implications for one of the greatest human adversaries: cancer. There are over 100 discrete forms of cancer, each with multiple causes (*National Cancer Institute, 2020*). Numerous studies have found that DCLK1 is upregulated and acts as an oncogene in a range of cancers including pancreatic, colorectal, gastric, bladder, and breast cancer (*Li and Bellows, 2013*; *Meng et al., 2013*; *Qu et al., 2015*; *Liu et al., 2016*; *Fan et al., 2017*; *Kadletz et al., 2017*; *Jiang et al., 2018*; *Zhang et al., 2017*). Due to the emerging body of evidence implicating DCLK1 in tumorigenesis, the protein appears to be a promising target for not just one, but for several types of cancers (*Westphalen et al., 2017*; *Weygant et al., 2014*; *Ferguson et al., 2020*). However, the complex intramolecular mechanism of DCLK1 must be thoroughly dissected before the field will be able to develop therapeutically effective drugs. For example, in light of our work, developing kinase inhibitors may not prove to be an effective means of controlling DCLK1's microtubule-binding functions, because WT and kinase-dead DCLK1 bind with similar affinities to microtubules, and prior studies have shown that they stimulate tubulin polymerization to similar extents (*Patel et al., 2016*). Future studies on the biological functions of DCLK1 microtubule-binding and kinase activity during

the initiation and progression of cancer cell proliferation and migration will provide fundamental insights into how DCLK1 contributes to this malady and how it can be adequately targeted.

# Materials and methods

## Key resources table

| Reagent type (species) or resource | Designation | Source or reference | Identifiers | Additional information |
|---|---|---|---|---|
| Antibody | Mouse anti-strep | Thermofisher | NBP243719 | |
| Antibody | Rabbit anti-thiophosphate ester | Abcam | ab133473 | |
| Antibody | Alexa 680 goat anti-mouse | Thermofisher | A28183 | |
| Antibody | Dylight 800 goat anti-rabbit | Rockland Labs | 611-145-002 | |
| Strain, strain background | *Escherichia coli* (BL21DE3) | Agilent | 200131 | |
| Strain, strain background | *Escherichia coli* (XL10Gold) | Agilent | 200314 | |
| Chemical compound, drug | Biotinylated poly(L-lysine)-[g]-poly(ethylene-glycol) (PLL-PEG-Biotin) | SuSoS AG | PLL(20)-G[3.5]-PEG(2)/PEG(3.4)-biotin(50%) | |
| Chemical compound, drug | Streptavidin | Thermofisher | 21135 | |
| Chemical compound, drug | Trolox (6-hydroxy-2,5,6,7,8-tetramethylchroman-2-carbonsaure, 97%) | Acros | AC218940050 | |
| Chemical compound, drug | 3,4-Dihydroxybenzoic Acid (protocatechuic acid) | Sigma-Aldrich | 37580 | |
| Chemical compound, drug | Protocatchuate 3,4-Dioxygenase from *Pseudomonas* sp. | Sigma-Aldrich | P8279 | |
| Chemical compound, drug | κ-caesin from bovine milk | Sigma-Aldrich | C0406 | |
| Chemical compound, drug | Pierce Bovine Serum Albumin, Biotinylated | Thermofisher | 209130 | |
| Chemical compound, drug | Paclitaxel | Sigma-Aldrich | T7402 | |
| Chemical compound, drug | Pluronic F-157 | Sigma-Aldrich | P2443 | |
| Other | Glass cover slides (18x18-1.5) | Fisher | 12-541A | |
| Other | Superfrost Microscope slides | Fisher | 12-550-143 | |
| Chemical compound, drug | Adenosine 5'-triphosphate disodium salt hydrate | Sigma-Aldrich | A2383 | |
| Chemical compound, drug | Guanosine 5'-triphosphate sodium salt hydrate | Sigma-Aldrich | G8877 | |
| Chemical compound, drug | Bovine Serum Albumin | Sigma-Aldrich | A2058 | |
| Chemical compound, drug | Casein | Sigma-Aldrich | C7078 | |
| Chemical compound, drug | Nonidet P 40 substitute (NP-40) | Sigma-Aldrich | 74385 | |

*Continued on next page*

*Continued*

| Reagent type (species) or resource | Designation | Source or reference | Identifiers | Additional information |
|---|---|---|---|---|
| Chemical compound, drug | PDMS (polydimethylsiloxane, sylgard 184) | Sigma-Aldrich | 761036 | |
| Chemical compound, drug | DNAseI | NEB | M0303L | |
| Chemical compound, drug | Streptactin Superflow resin | Qiagen | 30002 | |
| Chemical compound, drug | Streptactin XT Superflow resin | IBA | 2-4010-025 | |
| Chemical compound, drug | d-Desthiobiotin | Sigma | D1411 | |
| Chemical compound, drug | D-biotin | CHEM-IMPEX | #00033 | |
| Chemical compound, drug | ATPγS | Thermofisher | 11912025 MG | |
| Chemical compound, drug | p-Nitrobenzyl mesylate | Abcam | ab138910 | |
| Other | Phos-tag gels | Wako | 192–18001 | |
| Chemical compound, drug | GMPCPP | Jena Biosciences | NU-405 | |
| Recombinant protein | Calpain | Sigma | C6108 | |
| Recombinant DNA | Mouse DCLK1 | Transomics | #BC133685 | |
| Recombinant DNA | Lambda phosphatase | Addgene | 79748 | |
| Software, algorithm | FIJI | *Schindelin et al., 2012* | https://Fiji.sc/ | |
| Software, algorithm | GraphPad Prism | GraphPad | https://www.graphpad.com/scientific-software/prism/ | |
| Software, algorithm | μManager | *Edelstein et al., 2010* | https://micro-manager.org/ | |

## Molecular biology

The cDNAs (complementary DNA) used for protein expression in this study were as follows: mouse DCLK1 (Transomic, BC133685) and Lambda phosphatase (Addgene, 79748, RRID:Addgene_79748). DCLK1 proteins were cloned in frame using Gibson cloning into a pET28 vector with an N-terminal strepII-Tag and a superfolder GFP (sfGFP) cassette. λPP protein was cloned in frame using Gibson cloning into a pET11 vector with an N-terminal red fluorescent protein (RFP) cassette. The mouse DCLK1 (1–740) used in this study has the same amino acid boundaries as the human DCLK1 (UniPort O15075, isoform 2) modeled in the figures. There are only 11 differences between the human and the mouse sequences described in this paper: residues 172, 290, 294, 346, 357, 375, 408, 481, 625, 631, and 676; therefore, all of the residues that we mutate are conserved between the two sequences used in this paper. The modeling of the domains and motifs of human DCLK1 (UniProt O15075) that are conserved in the mouse DCLK1 used in this study is based on *Burgess and Reiner, 2001*; *Nagamine et al., 2011*. Models were based on the DC1 domain from 1mg4 (*Kim et al., 2003a*), the DC2 domain by homology to DCX-DC2 (5ip4; *Burger et al., 2016*), and the kinase domain (5jzj; *Patel et al., 2016*). We used IUPRED2A (*Mészáros et al., 2018*) to determine the disorder prediction score.

## Protein expression and purification

Tubulin was isolated from porcine brain using the high-molarity Piperazine-1,4-bis(2-ethanesulfonic acid) (PIPES) procedure as previously described (*Castoldi and Popov, 2003*). For bacterial expression of all sfGFP-DCLK1 variants, BL21 cells were grown at 37℃ until an ~OD of 0.6 was reached and protein expression was induced with 0.1 mM isopropyl-beta-D-thiogalactopyranoside (IPTG). For DCLK1 variants co-expressed with RFP-lambda phosphatase, BL21

cells were co-transformed with sfGFP-DCLK1 and RFP-lambda phosphatase and grown using the same protocol. Cells were grown overnight at 18°C, harvested, and frozen. Cell pellets were resuspended in lysis buffer (50 mM Tris, pH 8.0, 150 mM K acetate, 2 mM Mg acetate, 1 mM ethylene glycol tetraacetic acid (EGTA), 10% glycerol) with protease inhibitor cocktail (Roche), 1 mM dithiothreitol (DTT), 1 mM phenylmethylsulfonyl fluoride (PMSF), and DNAseI. Cells were then passed through an Emulsiflex press and cleared by centrifugation at 23,000 x*g* for 20 min. Clarified lysate from bacterial expression was passed over a column with Streptactin XT Superflow resin (Qiagen). After incubation, the column was washed with four column volumes of lysis buffer, then the bound proteins were eluted with 50 mM D-biotin (CHEM-IMPEX) in lysis buffer (pH 8.5). Eluted proteins were concentrated on Amicon concentrators and passed through a HiTrap Q HP anion exchange chromatography column in lysis buffer using a Bio-Rad NGC system. Peak fractions were collected, concentrated, and flash frozen in liquid nitrogen (LN$_2$). Protein concentration was determined by measuring the absorbance of the fluorescent protein tag and calculated using the molar extinction coefficient of the tag. The resulting preparations were analyzed by sodium dodecyl sulfate-polyacrylamide gel electrophoresis (SDS-PAGE).

## Autophosphorylation assays

For autophosphorylation assays to determine an intra- vs inter-molecular mechanism, 500 nM of WT and/or kinase-dead (D511N) DCLK1 proteins was incubated in the absence or presence of 0.5 mM ATPγS (Fisher 11912025 MG) in assay buffer containing 50 mM Tris, pH 8.5, 50 mM K acetate, 2 mM Mg acetate, 1 mM EGTA, and 10% glycerol, supplemented with 1 mM DTT and 1 mM PMSF for 30 min at 37°C. To determine if DCLK1-WT trans-phosphorylates DCLK1-D511N, we cleaved the strepII-sfGFP tag off the DCLK1-WT protein using Tobacco Etch Virus (TEV) protease and subsequently subjected the DCLK1-WT to gel filtration to separate it from the protease. All samples were quenched with 0.1 mM ethylenediaminetetraacetic acid (EDTA), then incubated at room temperature with 2.5 mM p-Nitrobenzyl mesylate (Abcam ab138910) for ~1 hr. All samples were then run on an SDS-PAGE gel and transferred to a polyvinylidene difluoride (PVDF) membrane using an iBlot2 at 25V for 7 min. The membrane was immunoblotted with primary antibodies mouse anti-strep (1:2500, Fisher NBP243719) and rabbit anti-thiophosphate ester (1:2000, Abcam ab133473, RRID:AB_2737094), washed, incubated with secondary antibodies Alexa 680 goat anti-mouse (1:10,000, Fisher A28183, RRID:AB_2536167) and Dylight 800 goat anti-rabbit (1:10,000, Rockland labs 611-145-002, RRID:AB_1660964), washed, and then imaged on a LiCor Odyssey.

For autophosphorylation assays using the Phos-tag gel system, assays were performed using ATP in assay buffer containing 50 mM Tris, pH 8.5, 50 mM K acetate, 2 mM Mg acetate, 1 mM EGTA, and 10% glycerol, supplemented with 1 mM DTT and 1 mM PMSF. The samples were incubated at room temperature for 30 or 60 min in the presence of assay buffer with 500 nM DCLK1 and 2 mM ATP. Non-phosphorylated samples included assay buffer and 500 nM DCLK1. All samples were run on a Phos-tag gel and analyzed as described below.

## Phos-tag gel assays

Purified proteins were separated using Phos-tag gel technology (Wako, Phos-tag AAL-107). Samples were either expressed alone in BL21 cells and purified, treated with λPP protein after purification at 30°C for 1 hr (New England BioLabs, P0753L) or co-expressed in BL21 cells with RFP-lambda phosphatase. Purified protein samples were incubated in the presence or absence of ATP in kinase assay protein buffer (50 mM Tris, pH 8.5, 50 mM K acetate, 2 mM Mg acetate, 1 mM EGTA, 10% glycerol) with 1 mM DTT and 1 mM PMSF for 30 or 60 min at room temperature. Phos-tag SDS-PAGE was performed with pre-cast 7.5% polyacrylamide gels containing 50 μM Phos-tag acrylamide with MnCl$_2$ (Wako, 192–18001). Electrophoresis was completed at 180 v for 90 min and the gel was stained with Coomassie blue. The stained gel was imaged using a GelDoc (BioRad) and the band intensity was quantified using ImageJ to draw a box over both the highest band and the lowest band in each lane. The measure of percent of protein that was phosphorylated was generated by dividing the intensity value of the highest band by the total intensity from the sum of the highest and lowest bands.

## Co-sedimentation assays

Co-sedimentation assays were performed as previously described (*Monroy et al., 2018*). Microtubules were prepared by polymerizing 25 mg/ml of porcine tubulin in assembly buffer (BRB80 buffer supplemented with 1 mM GTP, 1 mM DTT) at 37°C for 15 min, then a final concentration of 20 μM taxol was added to the solution, which was incubated at 37°C for an additional 15 min. Microtubules were pelleted over a 25% sucrose cushion at 100,000 ×$g$ at 25°C for 10 min, then resuspended in BRB80 buffer with 1 mM DTT and 10 μM taxol. SfGFP-DCLK1 proteins were first centrifuged at 100,000 x$g$ at 4°C for 10 min, then DCLK1 proteins were incubated in the presence or absence of 2 mM ATP at 25°C for 30 min in assay buffer (50 mM Tris, pH 8, 150 mM K acetate, 2 mM Mg acetate, 1 mM EGTA, 10% glycerol) supplemented with 1 mM DTT, 10 μM taxol, and 0.01 mg/ml bovine serum albumin (BSA). Binding reactions were then performed by mixing 500 nM of the sfGFP-DCLK1 protein with 2 μM microtubules in assay buffer and incubated at 25°C for 20 min. The mixtures were then pelleted at 90,000 x$g$ at 25°C for 10 min. Supernatant and pellet fractions were recovered, resuspended in sample buffer, and analyzed by SDS-PAGE. Protein band intensities were quantified using ImageJ.

## TIRF microscopy

For TIRF-M experiments, a mixture of native tubulin, biotin-tubulin, and fluorescent-tubulin purified from porcine brain (~10:1:1 ratio) was assembled in BRB80 buffer (80 mM PIPES, 1 mM MgCl$_2$, 1 mM EGTA, pH 6.8 with KOH) with 1 mM GTP for 15 min at 37°C, then polymerized microtubules were stabilized with 20 μM taxol. Microtubules were pelleted over a 25% sucrose cushion in BRB80 buffer to remove unpolymerized tubulin. Flow chambers containing immobilized microtubules were assembled as described (*McKenney et al., 2014*). Imaging was performed on a Nikon Eclipse TE200-E microscope equipped with an Andor iXon EM CCD camera, a X100, 1.49 NA objective, four laser lines (405, 491, 568, and 647 nm), and Micro-Manager software (*Edelstein et al., 2010*). All experiments were performed in assay buffer (60 mM 4- (2-hydroxyethyl) -1-piperazineethanesulfonic acid (HEPES), pH 7.4, 50 mM K acetate, 2 mM Mg acetate, 1 mM EGTA, and 10% glycerol) supplemented with 0.1 mg/ml biotin-BSA, 0.5% Pluronic F-168, and 0.2 mg/ml κ-casein (Sigma) and 10 μM taxol. When examining the effects of autophosphorylation on microtubule binding, purified DCLK1 proteins were incubated in the presence or absence of 2 mM ATP in the above assay buffer for 30 min at 25°C, then flowed into a chamber containing microtubules that were immobilized to the glass surface. The samples were treated exactly in the same manner whether or not they had ATP.

For imaging DCLK1 binding to non-taxol-stabilized microtubules, GMPCPP (Guanosine-5'-[(α,β)-methyleno]triphosphate) seeds were made from a mixture of 647-tubulin, biotin-tubulin, and unlabeled tubulin that was diluted to a final tubulin concentration of 30 μM in BRB80 + 1 mM DTT. The mixture was then incubated with 1 mM GMPCPP (Jena Biosciences, NU-405) at 37°C for 20 min, then spun through a 25% sucrose cushion for 10 min at 50,000 x$g$ at 37°C. The pellet was resuspended in BRB80 + 1 mM DTT, then the GMPCPP seeds were diluted 1:100 in assay buffer (BRB80 supplemented with 0.1 mg/ml biotin-BSA, 10% Pluronic F-168, and 0.2 mg/ml κ-casein) for the experiment. Prior to imaging DCLK1 proteins, 100 nM of DCLK1 was incubated in assay buffer in the absence or presence of 2 mM ATP for 30 min at 25°C. The proteins were then diluted to a final concentration of 10 nM in assay buffer that also contained 10 uM of a mixture of 405-tubulin and unlabeled tubulin and 2 mM GTP. The entire mixture was passed into the chamber containing GMPCPP seeds affixed to the coverslip, incubated for 5 min to allow for polymerization, then images were taken. For all saturation curves, a concentration series was performed for each protein. For fluorescence intensity analysis, ImageJ was used to draw a line across the microtubule of the DCLK1 channel and the integrated density was measured. The line was then moved adjacent to the microtubule of interest and the local background was recorded. The background value was then subtracted from the value of interest to give a corrected intensity measurement. The fluorescence intensity data were fit with a one-site-binding hyperbola equation to derive the K$_D$ for each DCLK1 variant. For our power of analysis, we decided to perform each experiment at least twice using two different protein preparations. In addition, we decided to analyze 50–100 microtubules from 5 to 10 images to ensure reproducibility.

## Mass spectrometry

Samples were prepared for mass spectrometry analysis by incubating each DCLK1 variant with ATP for time periods ranging from 5 to 15 min. The reaction was quenched with 10 mM EDTA. Protein of interest was first reduced at 56°C for 45 min in 5.5 mM DTT, followed by alkylation for 1 hr in the dark with iodoacetamide added to a final concentration of 10 mM. Trypsin was added at a final enzyme:substrate mass ratio of 1:50 and digestion carried out overnight at 37°C. The reaction was quenched by flash freezing in $LN_2$, and the digest was lyophilized. Digest was reconstituted in 0.1% trifluoroacetic acid with 10% acetonitrile prior to injection.

The mass spectrometry instrument used to analyze the samples was a Xevo G2 QTof coupled to a nanoAcquity UPLC system (Waters, Milford, MA). Samples were loaded onto a C18 Waters Trizaic nanotile of 85 um × 100 mm; 1.7 μm (Waters, Milford, MA). The column temperature was set to 45°C with a flow rate of 0.45 ml/min. The mobile phase consisted of A (water containing 0.1% formic acid) and B (acetonitrile containing 0.1% formic acid). A linear gradient elution program was used: 0–40 min, 3–40% (B); 40–42 min, 40–85% (B); 42–46 min, 85% (B); 46–48 min, 85–3% (B); 48–60 min, 3% (B).

Mass spectrometry data were recorded for 60 min for each run and controlled by MassLynx 4.1 (Waters, Milford, MA). Acquisition mode was set to positive polarity under resolution mode. Mass range was set from 50 to 2000 Da. Capillary voltage was 3.5 kV, with sampling cone at 25 V and extraction cone at 2.5 V. Source temperature was held at 110°C. Cone gas was set to 25 l/h, nano flow gas at 0.10 bar, and desolvation gas at 1200 l/h. Leucine–enkephalin at 720 pmol/ul (Waters, Milford, MA) was used as the lock mass ion at m/z 556.2771 and introduced at 1 ul/min at 45-s intervals with a three-scan average and mass window of +/- 0.5 Da. The $MS^e$ data were acquired using two scan functions corresponding to low energy for function 1 and high energy for function 2. Function 1 had collision energy at 6 V and function 2 had a collision energy ramp of 18–42 V.

RAW $MS^e$ files were processed using Protein Lynx Global Server (PLGS) version 2.5.3 (Waters, Milford, MA). Processing parameters consisted of a low-energy threshold set at 200.0 counts, an elevated energy threshold set at 25.0 counts, and an intensity threshold set at 1500 counts. The databank used was derived from humans. Searches were performed with trypsin specificity and allowed for two missed cleavages. Possible structure modifications included for consideration were methionine oxidation, carbamidomethylation of cysteine, and phosphorylation of serine, threonine, or tyrosine.

For viewing, ProteinLynx Global SERVER (PLGS) search results were exported in Scaffold v4.4.6 (Proteome Software Inc, Portland, OR).

## Sucrose gradients

Three-step sucrose gradients were prepared in centrifuge tubes using 250 μl steps of 3, 6, and 9% sucrose in protein buffer and allowed to sit overnight at 4°C. The next morning, 100 μl of ~600 nM of each DCLK1 variant was layered on top of the sucrose gradient, which was centrifuged at 50,000 rpm for 4 hr at 4°C using a TLS55 rotor and an Optima MAX-XP Ultracentrifuge (Beckman Coulter). The sample was physically fractionated into 10 fractions of 85 ul each, which were run on an SDS-PAGE gel to assess protein sedimentation location. For quantification of protein levels in each fraction, ImageJ was used to draw a box of consistent size over the band in each lane and total intensity within the box was measured. The intensity of the band in each lane was divided by the total intensity of all boxes combined to produce the percent of protein in each fraction.

## Calpain cleavage assay

Calpain cleavage assays were performed in assay buffer containing 20 mM HEPES, pH 7.2, 100 mM KCl, and 2 mM $CaCl_2$. The DCLK1 proteins were initially purified in the presence of λPP that was subsequently removed. 750 nM of DCLK1 protein was first incubated in assay buffer in the absence or presence of 2 mM ATP for 30 min at 25°C, then incubated at 30°C for 15 min in the presence of 0.62 μg human calpain 1 (Sigma, C6108), and analyzed by SDS-PAGE.

## Statistical analysis

Statistical tests were performed with two-tailed unpaired Student's t-test or one-way analysis of variance (ANOVA) with Bonferroni post-hoc correction. Unless otherwise stated, all data were analyzed

manually using ImageJ (FIJI). Graphs were created using Graphpad Prism and statistical tests were performed using this program. All variances given represent standard deviation. The statistical details of each experiment can be found in the figure legends.

## Acknowledgements

We thank Richard McKenney and members of the Ori-McKenney and McKenney labs for reading the manuscript and providing feedback.

## Additional information

### Competing interests

Kassandra M Ori-McKenney: Reviewing editor, *eLife*. The other authors declare that no competing interests exist.

### Funding

| Funder | Grant reference number | Author |
|---|---|---|
| National Institutes of Health | 1R35GM133688 | Kassandra M Ori-McKenney |
| Pew Charitable Trusts | A19-0406 | Kassandra M Ori-McKenney |

The funders had no role in study design, data collection and interpretation, or the decision to submit the work for publication.

### Author contributions

Regina L Agulto, Data curation, Formal analysis, Validation, Investigation, Visualization, Methodology, Writing - review and editing; Melissa M Rogers, Conceptualization, Resources, Formal analysis, Validation, Investigation, Visualization, Methodology, Writing - original draft, Writing - review and editing; Tracy C Tan, Formal analysis, Validation, Investigation, Methodology; Amrita Ramkumar, Formal analysis, Investigation; Ashlyn M Downing, Hannah Bodin, Julia Castro, Resources; Dan W Nowakowski, Data curation; Kassandra M Ori-McKenney, Conceptualization, Supervision, Funding acquisition, Visualization, Methodology, Writing - original draft, Writing - review and editing

### Author ORCIDs

Kassandra M Ori-McKenney ⓘD https://orcid.org/0000-0003-2051-2495

### Decision letter and Author response

Decision letter https://doi.org/10.7554/eLife.60126.sa1
Author response https://doi.org/10.7554/eLife.60126.sa2

## Additional files

### Supplementary files

• Transparent reporting form

### Data availability

All data generated or analyzed during this study are included in the manuscript or as source files.

The following datasets were generated:

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
