## [Decision Letter]

**Acceptance summary:**

The authors studied DCLK1, a member of the doublecortin family of microtubule-associated proteins, with roles in brain development and cancer. The uncover previously unknown mechanisms of regulation via auto-phosphorylation. DCLK1 auto-phosphorylation of its C-terminal tail suppresses its kinase activity. This, in turn, reduces hyperphosphorylation of its doublecortin domains (particularly DC2) increasing the protein's ability to bind microtubules.

**Decision letter after peer review:**

Thank you for submitting your article "Autoregulatory control of microtubule binding in the oncogene, doublecortin-like kinase 1" for consideration by *eLife*. Your article has been reviewed by 3 peer reviewers, and the evaluation has been overseen by a Reviewing Editor and Piali Sengupta as the Senior Editor. The reviewers have opted to remain anonymous.

The reviewers have discussed the reviews with one another and the Reviewing Editor has drafted this decision to help you prepare a revised submission.

Summary:

The authors describe the control of microtubule binding by autophosphorylation of doublecortin-like kinase 1. They show that the C-terminal tail controls the activity of the kinase domain and removal of the C-terminus or mutation of an autophosphorylation site results in hyperphosphorylation of the two microtubule binding doublecortin domains (DC1 and DC2). Mutation of a number of these phosphorylation sites in DC2 significantly reduces microtubule binding. The data suggest that autophosphorylation of the C-terminal tail is required to suppress kinase activity, which in turn increases microtubule binding.

Essential revisions:

1) There is a key question related to the physiological relevance of the work that remains unanswered: Doublecortin domains have been shown to be sensitive to protofilament number and lattice expansion/compaction with straight, taxol-stabilised microtubules not being a preferred substrate for doublecortin to bind to, nor is it a physiologically relevant substrate. It is recommended that some binding studies are also performed with MTs not stabilised with taxol as this could make quite a difference to DC domain binding.

2) A major technical issue is the huge difference in background labelling in the TIRF-M data in all the figures. There was concern about the background labelling being higher in those samples in which microtubule decoration is higher and it calls into question whether the data are reliable, the inputs/free protein concentrations are indeed comparable and the microtubule binding specific or some constructs just more prone to crash out on the surface. Given that affinity measurements using TIRF are the sole assay used to compare microtubule binding, this is a key issue. The protein band intensities are also different on the protein gel provided, which isn't adding confidence that protein preparations have been normalised for affinity measurements.

A reviewer suggested the background of microtubules could be used as proxy for protein present in solution – of course after subtracting camera offset etc. There should be a clear linear increase with concentration and different mutant proteins should have comparable intensities. The averages of free and bound should be reported for each repeat rather than just pooled data. Another way to provide support for the binding affinity data is to perform microtubule pelleting assays and derive bound and unbound fractions from pellet/supernatant.

3) We noted that some data are based on two repeats and statistics seems to be based on the number of microtubule analysed rather than the number of experimental repeats. Given the potential difficulties in ensuring a comparable free concentration of proteins in the assay chamber, repeat assays are the crucial variation here, ideally based on different protein preparations.

4) Figure 1: When proteins are treated with phosphatase to remove charged phosphates, this causes a change in the π of the protein that might lead to the protein to precipitate. It would be great if the authors could provide some quality control of the samples used for TIRF. An easy way to do this would be to run the phosphatase treated protein over size exclusion chromatography to check that they have not aggregated.The phosphatase treatment data in Figure 1D-F miss an important control, D511N + PP. This control would exclude other phosphorylation sources playing a role here.

5) Figure 4/5: The level of expression of the T688A and T687A/T688A and DCLK1 ∆C4A protein seem to be a lot lower than other constructs. This may indicate a lack of stability of the protein expressed. Considering the data presented are highly dependent on the amount of protein used (3nM, 12nM and 48nM) it is critical then to know that the protein added in the assay is "functional". The authors should demonstrate that each mutant has a similar elution profile on size exclusion chromatography to the WT protein to ensure that the mutants are similarly folded. Only then conclusions can be drawn on the impact of the mutations.

---

## [Author Response]

Essential revisions:1) There is a key question related to the physiological relevance of the work that remains unanswered: Doublecortin domains have been shown to be sensitive to protofilament number and lattice expansion/compaction with straight, taxol-stabilised microtubules not being a preferred substrate for doublecortin to bind to, nor is it a physiologically relevant substrate. It is recommended that some binding studies are also performed with MTs not stabilised with taxol as this could make quite a difference to DC domain binding.

We thank the reviewers for this suggestion. We have performed the binding assays for WT and DC DCLK1 in the presence and absence of ATP on GDP lattices grown off of GMPCPP seeds (Figure 2F-G), and we observe similar behaviors on a true GDP lattice as we do with taxolstabilized lattices. We have performed comparison assays between taxol-stabilized and nonstabilized GDP lattices in the past with a number of different MAPs (MAP7, MAP9, MAP1B, etc.) and have also not observed a difference. Thus, although taxol is not physiologically relevant, we do believe that it can be used to study the basic biophysical binding properties of these MAPs. Also, recent work from the Akhmanova lab has shown that taxol-stabilized microtubules predominantly contain 13 protofilaments (Rai et al., BioRxiv 2021), indicating that the binding site for a doublecortin domain should be preserved for this family of MAPs.

2) A major technical issue is the huge difference in background labelling in the TIRF-M data in all the figures. There was concern about the background labelling being higher in those samples in which microtubule decoration is higher and it calls into question whether the data are reliable, the inputs/free protein concentrations are indeed comparable and the microtubule binding specific or some constructs just more prone to crash out on the surface. Given that affinity measurements using TIRF are the sole assay used to compare microtubule binding, this is a key issue. The protein band intensities are also different on the protein gel provided, which isn't adding confidence that protein preparations have been normalised for affinity measurements.A reviewer suggested the background of microtubules could be used as proxy for protein present in solution – of course after subtracting camera offset etc. There should be a clear linear increase with concentration and different mutant proteins should have comparable intensities. The averages of free and bound should be reported for each repeat rather than just pooled data. Another way to provide support for the binding affinity data is to perform microtubule pelleting assays and derive bound and unbound fractions from pellet/supernatant.

We thank the reviewers for pointing this out, and we agree that the quality of images at higher concentrations (> 25 nM) did not meet our standards. It is important to point out that this did not indicate that some proteins were “prone to crash out,” because the background labeling was the same at high concentrations of all of our proteins, not just the mutants. In addition, Figure 1figure supplement 1 was only meant to show purity, not equal loading of the proteins. We repeated all of these assays with both prior preps and with multiple new preparations of proteins and determined the primary reason behind this background labeling was the coverslips. All of the data are consistent, as repeating all of these assays produced the same results, but the data we generated with cleaner coverslips is of higher quality and all of the images have been replaced in the manuscript. There is still some background labeling at higher concentrations, but far less than in the prior datasets, and we measure and take this background into consideration in our intensity measurements. In addition, we have performed microtubule co-pelleting of WT vs. DC (Figure 1figure supplement 2D) and WT vs. T688A (Figure 5G) in the absence or presence of ATP, as suggested, and observe similar effects of aberrant autophosphorylation on microtubule binding in this bulk assay. We thank the reviewers for this suggestion for strengthening our paper.

3) We noted that some data are based on two repeats and statistics seems to be based on the number of microtubule analysed rather than the number of experimental repeats. Given the potential difficulties in ensuring a comparable free concentration of proteins in the assay chamber, repeat assays are the crucial variation here, ideally based on different protein preparations.

We agree with the reviewers and we have increased the number of independent trials for our experiments and performed statistics based on the number of independent experiments. We also performed all experiments with at least two different protein preparations for each construct. All of this information has been added or updated in the figure legends and methods.

4) Figure 1: When proteins are treated with phosphatase to remove charged phosphates, this causes a change in the π of the protein that might lead to the protein to precipitate. It would be great if the authors could provide some quality control of the samples used for TIRF. An easy way to do this would be to run the phosphatase treated protein over size exclusion chromatography to check that they have not aggregated.The phosphatase treatment data in Figure 1D-F miss an important control, D511N + PP. This control would exclude other phosphorylation sources playing a role here.

We thank the reviewers for this exciting idea! We now include the anion exchange elution chromatograms for the normally expressed WT DCLK1 that has either been treated or not treated with lamba phosphatase and the normally expressed DC DCLK1 that has either been treated or not treated with lamba phosphatase (Figure 1—figure supplement 1C). All proteins, regardless of treatment, elute in a homogenous peak. In addition, the shifts observed between the nonphosphorylated and phosphorylated proteins are consistent with a change in net negative charge. We have also included a PhosTag gel of D511N either treated or not treated with λ phosphatase in Figure 1—figure supplement 2C.

5) Figure 4/5: The level of expression of the T688A and T687A/T688A and DCLK1 ∆C4A protein seem to be a lot lower than other constructs. This may indicate a lack of stability of the protein expressed. Considering the data presented are highly dependent on the amount of protein used (3nM, 12nM and 48nM) it is critical then to know that the protein added in the assay is "functional". The authors should demonstrate that each mutant has a similar elution profile on size exclusion chromatography to the WT protein to ensure that the mutants are similarly folded. Only then conclusions can be drawn on the impact of the mutations.

We thank the reviewers for this suggestion. We have updated the protein prep gels and we also provide the anion exchange elution chromatograms for DCLK1-WT compared to T687A, T688A, and D511N, and DCLK1-DC compared to ∆C_DC1-4A_ and ∆C_DC2-4A_ showing that all proteins elute in a homogenous peak at approximately the same ionic strength revealing that they are similarly folded (Figure 1—figure supplement 1B).